# SYMMETRIC-APL ACTIVATIONS: TRAINING INSIGHTS AND ROBUSTNESS TO ADVERSARIAL ATTACKS

## ABSTRACT

Deep neural networks with learnable activation functions have shown superior performance over deep neural networks with fixed activation functions for many different problems. The adaptability of learnable activation functions adds expressive power to the model which results in better performance. Here, we propose a new learnable activation function based on Adaptive Piecewise Linear units (APL), which 1) gives equal expressive power to both the positive and negative halves on the input space and 2) is able to approximate any zero-centered continuous non-linearity in a closed interval. We investigate how the shape of the Symmetric-APL function changes during training and perform ablation studies to gain insight into the reason behind these changes. We hypothesize that these activation functions go through two distinct stages: 1) adding gradient information and 2) adding expressive power. Finally, we show that the use of Symmetric-APL activations can significantly increase the robustness of deep neural networks to adversarial attacks. Our experiments on both black-box and open-box adversarial attacks show that commonly-used architectures, namely Lenet, Network-in-Network, and ResNet-18 can be up to 51% more resistant to adversarial fooling by only using the proposed activation functions instead of ReLUs.

## 1 INTRODUCTION

Deep neural networks (DNNs) are a series of linear transformations followed by point-wise non-linear functions. These non-linear functions are also referred to as activation functions. Activation functions directly influence the training dynamics and, therefore, affect the final performance of the DNN. For instance, activations such as Sigmoid and Tanh suffer from what is known as the "vanishing gradient" problem, when the inputs are in the saturation regions of these functions. This can adversely affect network performance and cause slow convergence. Rectified linear units (ReLU) (Nair & Hinton, 2010) have been shown to perform better than Sigmoid and Tanh in many scenarios. Instead of using a fixed activation function, one can learn a parameterized activation function to add expressive power to the model. Towards this goal, significant efforts have been made to introduce different learnable activation functions (Agostinelli et al., 2014; He et al., 2015; Ramachandran et al., 2017; Jin et al., 2016; Li et al., 2016). Although most of these adaptive activation functions outperform fixed activation functions in multiple tasks, there is very little insight into why.

In this paper, we propose the Symmetric Adaptive Piecewise Linear unit (Symmetric-APL; pronounced as Symmetric apple: ˈæp.əl) as a new adaptive activation function which builds upon the adaptive piecewise linear unit (APL) (Agostinelli et al., 2014). Similar to APL units, S-APL has multiple hinges which are connected by linear pieces. The S-APL activation is designed to give the same expressive power to both the positive and negative sides of the x-axis. We show that S-APL can approximate any continuous and zero-centered function, which encompasses many types of learned and fixed activations functions. Using this flexible learned activation function, we can then analyze how the shape changes during training and use ablation studies to investigate the role that these changes play in optimization. This helps us better understand learned activation functions and explain why they often provide increases in performance.

Finally, we evaluate the robustness of S-APL activated networks to adversarial attacks. Experiments are done using both black-box and open-box adversarial attacks where we observe S-APL equipped

networks are more robust to attacks than their ReLU counterparts. Experiments show that the robustness improves against differential evolution attacks (Su et al., 2019b) by (up to) 51% on Lenet , 47% on ResNet-18 (He et al., 2016), 43% on pure CNN, and 38% on Net in Net (Lin et al., 2013). For open-box adversarial attacks, the robustness also improves against FGSM and CW-L2 (Goodfellow et al., 2014) attack by (up to) 36%, 28%, and 25% for Lenet, pure CNN, and ResNet-18 architectures, respectively using CIFAR-10 dataset.

## 2 RELATED WORKS

ReLU and its variants such as Leaky-ReLU (Maas et al., 2013), P-ReLU (He et al., 2015), exponential linear units (ELU) (Clevert et al., 2015), and scaled exponential linear units (SELU) (Klambauer et al., 2017) are believed to solve the vanishing gradient problem and some are shown to better normalize the network. By changing the negative side of x-axis, these variants push the mean activations of each unit closer to zero and accelerate learning.

Some of the pioneering attempts to learn activations in a neural network can be found in Poli (1996), Weingaertner et al. (2002), and Khan et al. (2013) where the authors proposed novel approaches to learning the best activation function per neuron among a pool of candidate activations by employing genetic and evolutionary algorithms. Maxout (Goodfellow et al., 2013) has been introduced as an activation function aimed at enhancing the model averaging properties of dropout (Srivastava et al., 2014). However, not only is it limited to approximating convex functions, but it also requires a significant increase in parameters. Neural architecture search (Ramachandran et al., 2017) uses a combination of exhaustive and reinforcement learning based search. Although the authors discovered multiple novel activation functions, of which $f(x) = x * sigmoid(\beta x)$ performed slightly better than ReLU, the search space of activation functions is limited. S-ReLU (Jin et al., 2016) was proposed as an adaptive activation function which can mimic both convex and non-convex functions.

It is worth mentioning that, in Lin et al. (2013), the authors proposed the network in network approach where they replace activation functions in convolutional layers with small multi-layer perceptrons. Theoretically, due to Universal Approximation Theorem, this is the most expressive activation; however, it requires many more parameters.

## 3 ADAPTIVE PIECEWISE ACTIVATION FUNCTION

### 3.1 ADAPTIVE PIECEWISE LINEAR UNITS

Agostinelli et al. (2014) proposed an adaptive activation function which consists of multiple linear pieces. The formulation of this adaptive piecewise linear (APL) activation function is shown in Equation 1.

$$h_i(x) = max(0, x) + \sum_{s=1}^{S} a_i^s max(0, -x + b_i^s) \tag{1}$$

Where $h_i$ is the non-linear transformation for the hidden unit $i$ and $S$ is a hyperparameter that determines the number of hinges. $a_i$ is the slope of the linear piece and $b_i$ is the location of the hinge.

However, APL units are not zero-centered as they can have a non-zero output for a input of zero. This behavior provides no apparent beneficial purpose. Furthermore, APL units can only represent piecewise linear functions whose output $h_i(x) = x$ for $x > u$, for some scalar $u$. This significantly restricts the class of piecewise linear functions that APLs can express.

### 3.2 SYMMETRIC ADAPTIVE PIECEWISE LINEAR UNITS (APL: ˈæp.əl)

To overcome the shortcomings of APL activation function, we define a new adaptive piecewise linear activation function that is zero-centered and gives the same expressive power to the positive and negative half of the input space. Since S-APL shares the variables $a_+^s, b_+^s, a_-^s$, and $b_-^s$ among all

the neurons of a layer (e.i., $h_i(x, S)$ does not depend on $i$), it requires fewer parameters in compare to the plain APL. We formulate the activation $h_i$ of the hidden unit $i$ as the summation of symmetric hinge-shaped pieces.

$$h_i(x, S) = \sum_{s=1}^{S} a_+^s max(0, x - b_+^s) + \sum_{s=1}^{S} a_-^s max(0, -x - b_-^s) \qquad (2)$$

Similar to APL units, $S$ is a hyperparameter which determines the number of hinges (on each side of the x-axis). $a_+^s$ and $a_-^s$ control the slope of linear pieces in both the positive and negative sides of the x-axis, while $b_+^s, b_-^s \in [0, +\infty)$ determine the location of hinge $s$. The total number of additional parameters is $4S$ for a single layer. Because our networks are trained with batch normalization (Ioffe & Szegedy, 2015) before each S-APL layer, the distribution of input to the S-APL function will follow that of a normal distribution. Therefore, we fix the hinges to be a pre-determined number of standard deviations away from the mean, decreasing the number of parameters to $2S$.

Based on Equation 2, S-APL activation is zero-centered which leads to more stable statistical behavior during training. In case of using fixed hinges' locations for APL, due to increasing the number of parameters, S-APL offers more flexibility than the normal APL. However, this increase is of $O(constant)$ which is negligible. With the assumption of $a_+^s = a_-^s$ and $b_+^s = b_-^s$, since $h_i^s(x) = h_i^s(-x)$, S-APL allows neurons to activate symmetrically. As it is mentioned in Zhao & Griffin (2016), networks with symmetric activation functions are more robust to adversarial fooling. We will delve into this property in later sections.

S-APL can better handle the limits. It could be inferred from the definition that, in contrast to normal APL, increasing the absolute value of the input $x$ will also increase the number of terms affecting the $h_i(x)$. However, normal APL requires the rightmost linear section in all the component functions to have a unit slope and bias 0, which is not an appropriate constraint and undermines its representation ability.

The following theorem shows that S-APL can approximate any non-linear, zero-centered, and Lipschitz continuous function in a closed interval of real numbers. It is worth mentioning that APL was not flexible enough to do so.

**Theorem 3.1** *For any function $f : [A, B] \to \mathbb{R}$ and $\epsilon \in \mathbb{R}$, $\exists S \in \mathbb{N}$, where $|f(x) - \text{S-APL}(x)| \le \epsilon$, assuming:*

- *$A$ and $B$ are finite real numbers.*

- *$f$ is M-Lipschitz continuous.*

**Proof:** *We start by dividing the interval $[A, B]$ into $S$ equal sub-intervals $[A_i, B_i]$. In other words, we put $S$ equally distanced hinges in the interval $[A, B]$. So the length of each sub-interval will be $\frac{B-A}{S}$. We start approximating $f$ from the left most sub-interval which is $[A, A + \frac{B-A}{S}]$. The approximation of $f$ in that sub-interval will be a line which connects $f(A)$ to $f(A + \frac{B-A}{S})$. Due to M-Lipschitz continuity of $f$, for any $x \in [A, A + \frac{B-A}{S}]$:*

$$|f(x) - \text{S-APL}(x)| \le \begin{cases} |f(A) - f(A + \frac{B-A}{S}) + \frac{M(B-A)}{S}| & f(A) \le f(A + \frac{B-A}{S}), \\ |f(A + \frac{B-A}{S}) - f(A) + \frac{M(B-A)}{S}| & f(A) \ge f(A + \frac{B-A}{S}). \end{cases} \qquad (3)$$

*Again, due to M-Lipschitz continuity of $f$, we know that:*

$$|f(A) - f(A + \frac{B-A}{S})| \le \frac{M(B-A)}{S} \qquad (4)$$

*Using Equation 4, we can re-write the inequality 3 as follows:*

$$|f(x) - \text{S-APL}(x)| \le \frac{2M(B-A)}{S} \qquad (5)$$

*So, by choosing $S \ge \frac{2M(B-A)}{\epsilon}$, we satisfy that $|f(x) - \text{S-APL}(x)| \le \epsilon$.*

*in order to complete the proof, we need to show that this approximation can be applied to other sub-intervals of $[A, B]$ as well. We do so by induction as follows: Suppose that the approximation is done for sub-intervals $[A_1, B_1], ..., [A_i, B_i]$, now we prove that for sub-interval $[A_{i+1}, B_{i+1}]$ we can do the approximation which is connecting $f(A_{i+1})$ to $f(B_{i+1})$. The slope of S-APL in the sub-interval $[A_i, B_i]$ is $\sum_1^i a_+^i$ or $\sum_1^i a_-^i$ (depending on the sign of the sub-interval). However, the slope of S-APL in the sub-interval $[A_{i+1}, B_{i+1}]$ will be either $\sum_1^{i+1} a_+^i$ or $\sum_1^{i+1} a_-^i$. In both cases, the extra term $a_+^{i+1}$ or $a_-^{i+1}$ is not fixed and can change the slop to any arbitrary value. This fact plus the assumption of continuity of S-APL guarantee that $f(A_{i+1})$ can be connected to $f(B_{i+1})$ which was our proposed approximation. As we showed the approximation for the left most sub-interval, the induction is complete.* □

## 4 COMPARISON TO COMMONLY USED ACTIVATION FUNCTIONS

In order to show that S-APL unit is beneficial for deep neural networks, we compare it with well-known activation functions in different architectures. We employed five different neural networks trained on three different datasets, MNIST, CIFAR-10, and CIFAR-100. We used $S = 4$ and set the hinges at positions $x = -2.5, -2, -1, 0, +1, +2, +2.5$ (0 considered as two hinges). Slope of $a_1^{s^+}$ is initialized to 1 and the remaining slopes are initiazlied to 0. With this initialization, the initialized S-APL mimics the shape of ReLU.

The results of the experiments are shown in Table 1. The table shows that S-APL leads to a better validation accuracy in almost all architectures.

Table 1: ReLU, leaky-ReLU, PReLU, tanh, sigmoid, ELU, maxout (nine features), swish: $x.sigmoid(\beta x)$ and S-APL activations are compared in three different tasks. The performance of S-APL in comparison to the best of the other activations. For the sake of brevity, D-A refers to Data Augmentation. The values in the tables are error-rates and are reported in percentage.

| Activation | MNIST | CIFAR-10 | | CIFAR-100 | |
|---|---|---|---|---|---|
| | - | - | D-A | - | D-A |
| Lenet5 (best: PReLU) | 1.11 | 31.93 | 27.42 | 46.31 | 45.39 |
| Lenet5 (S-APL) | 1.03 | 30.83 | 27.02 | 45.74 | 44.81 |
| MLP (best: swish) | **1.55** | | | | |
| MLP (S-APL) | 1.71 | | | | |
| pure CNN (best: maxout) | | 12.01 | 10.92 | 35.02 | 35.33 |
| pure CNN (S-APL) | | 11.81 | 10.32 | 34.42 | 34.10 |
| ResNet-18 (best: PReLU) | | 9.41 | 9.02 | 20.14 | 19.60 |
| ResNet-18 (S-APL) | | 8.93 | 8.07 | 19.24 | 18.71 |
| EffectiveNet B0 (best: ReLU) | | 7.03 | 6.43 | 15.91 | 15.65 |
| EffectiveNet B0 (S-APL) | | **6.54** | **5.93** | **15.01** | **15.31** |
| Net-in-Net(ReLU) | | 10.41 | 8.81 | 33.72 | 31.63 |
| Net-in-Net(APL) | | 9.59 | 7.51 | 32.40 | 30.83 |
| Net-in-Net(S-APL) | | **9.41** | **7.20** | **32.12** | **30.33** |

## 5 HOW S-APL CHANGES DURING TRAINING AND POSSIBLE REASONS WHY

Figures 1 and 2 show how the shape of S-APL changes for each layer during training. From these figures, we can see that during the early stages of training, the shape of S-APL looks similar to that of leaky ReLU: having a negative output for $x < 0$ and a positive output for $x \geq 0$. During the later stages of training, the activation function has a positive output for both $x < 0$ and $x \geq 0$. In addition, the slope of the output decreases as the magnitude of $x$ increases.

We hypothesize that there are two distinct stages during the optimization of these learned activation functions. In the first stage, the function returns negative values for $x < 0$ and in the second stage,

the function returns positive values for $x < 0$. To investigate this hypothesis, we train S-APL under two different conditions where we: 1) force the output to only be negative for $x < 0$ (S-APL-negative) and 2) force the output to only be positive for $x < 0$ (S-APL-positive). We accomplish this by taking the absolute value of all $a^{s^-}$ and multiplying by $+1$ or $-1$.

Figure 2 shows that, although S-APL-positive has the ability to mimic the final learned shape of the S-APL function, it barely deviates from the ReLU initialization. This shows that the ability to give a negative output for negative inputs is crucial for S-APLs. Figure 2 shows that S-APL-negative does initially go negative in the 4th layer and then returns to the ReLU shape. To gain further insight, we compare the loss of these activation functions in Figure 3. We see that S-APL has a lower loss compared to S-APL-positive and S-APL-negative. Furthermore, S-APL-negative has a lower loss compared to S-APL-positive which has a lower loss than ReLU.

These results lead us to believe that it is possible that the first stage, S-APL-negative, adds gradient information by giving negative output for negative inputs. It is possible that S-APL-positive is not capable of doing so because giving positive outputs for negative inputs creates an identifiability issue. In other words, outputs for $x < 0$ and $x \geq 0$ look the same. However, we can see that this property is useful for the S-APL function in the later stages of training. It is possible that this is because the DNN has adapted its parameters such that outputs on both the negative and positive sides capture a similar concept, adding expressive power to the model. Furthermore, one cannot achieve the final S-APL shape without the first stage of having negative outputs for negative inputs. To conclude, it appears that the S-APL units first go through two stages: 1) adding gradient information and 2) adding expressive power and that the second stage depends on the first stage. This information can be used to inform the design of learnable activation functions and even DNN optimization techniques.

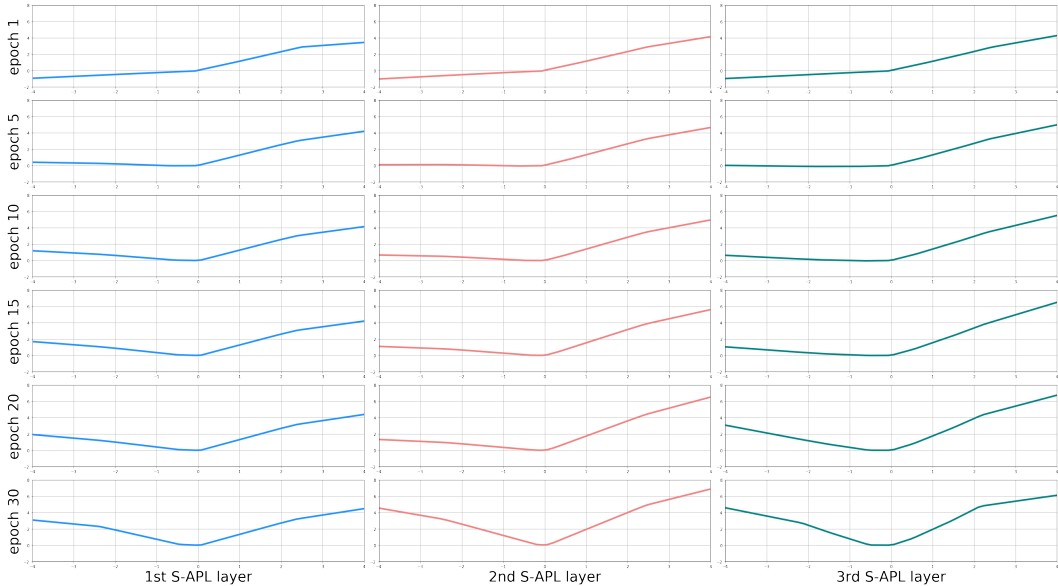

Figure 1: Shape of S-APL activation during training a simple network of MLPs on MNIST dataset.

## 6 ROBUSTNESS TO ADVERSARIAL ATTACKS

DNNs have been shown to be vulnerable to many types of adversarial attacks (Szegedy et al., 2013; Goodfellow et al., 2014). Research suggests that activation functions are a major cause of this vulnerability (Zantedeschi et al., 2017). Zhang et al. (2018) bounds a given activation function with a few linear and quadratic functions and allows it to tackle general activation functions. This adds up with applying a different activation for each neuron so the resulting network shows efficiency and robustness to adversarial foolings. Wang et al. (2018) proposed a data-dependent activation function and empirically shows its robustness to both black-box and gradient-based adversarial attacks. Other

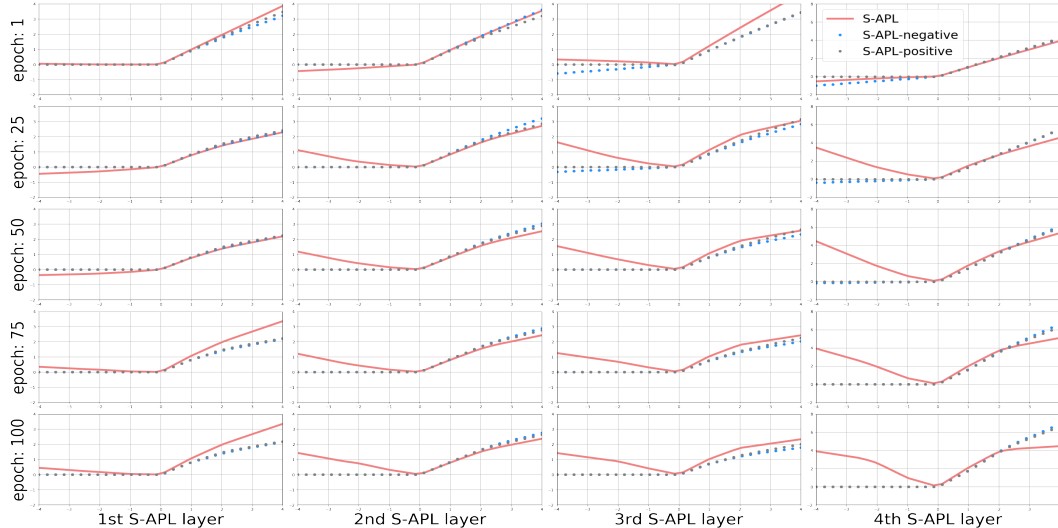

Figure 2: Shape of S-APL during training a Lenet5 architecture on CIFAR-10 dataset.

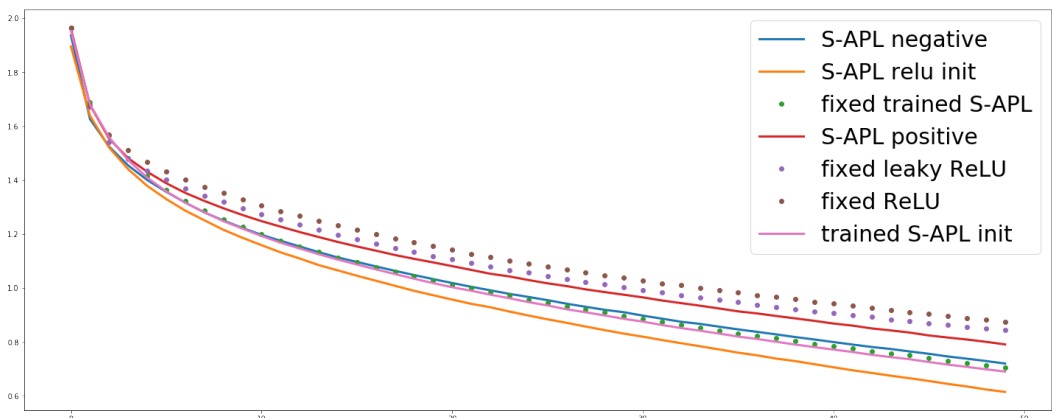

Figure 3: Training loss trajectory for different S-APL initializations compared to fixed ReLU and leaky ReLU.

studies such as Rakin et al. (2018), Dhillon et al. (2018), and Song et al. (2018) focused on other additional properties of activation such as quantization and pruning and showed they can improve the robustness of DNNs against adversarial examples.

Recently, Zhao & Griffin (2016) theoretically showed that DNNs with symmetric activations are less likely to get fooled. The authors proved that *"symmetric units suppress unusual signals of exceptional magnitude which result in robustness to adversarial fooling and higher expressibility."* Due to the symmetric convergence of S-APL units and also their hinges' locations ($b_i^+ = b_i^-$) which result in $h(x) = h(-x)$, S-APL units are capable of increasing the robustness of DNNs against adversarial attacks.

In this section, relying on the properties of S-APL such as data dependency, piecewise linearity, non-injective behavior, better handling of extreme values, and most importantly, symmetric shape, we show that adding S-APL to a DNNs greatly improves the robustness against adversarial attacks. This claim is verified through a wide range of experiments with the CIFAR-10 dataset under both black-box and open-box methods, including one-pixel-attack and Fast Gradient Sign Method.

Figure 4 provides an intuition for the robustness of S-APL activated network in comparison to ReLU activated ones. For each of the two networks, we take 100 random samples of frog and ship images and visualize the pre-softmax representations using tSNE visualization (Maaten & Hinton, 2008) in

Figure 4. As it is depicted, for S-APL activated networks, the two classes are better separated. We suspect that red samples such as those that are lying on the outer circle in the left plot of Figure 4 might be somewhere on the manifold of the data where the network is more susceptible. As one can see, there is no red point on the outer circle of the right plot which might be showing the robustness of S-APL networks. For other samples on the inner circle and the cloud next to circles, we can see better separation by the S-APL network.

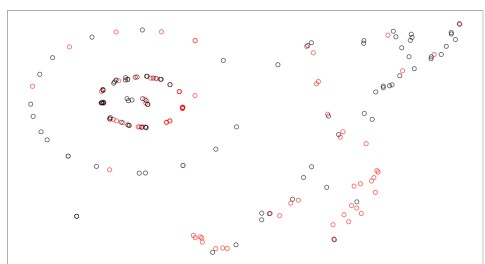 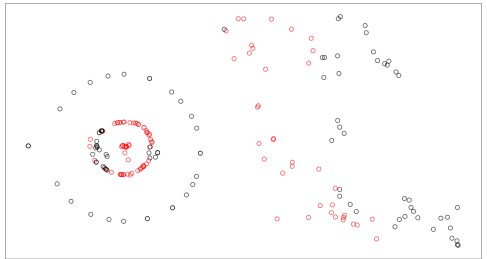

Figure 4: tSNE visualization of the pre-softmax layer's outputs. Left: a ReLU activated Lenet5 architecture trained on CIFAR-10. Right: same network trained with S-APL activation functions.

## 6.1 BLACK-BOX ADVERSARIAL ATTACK

For black-box attacks against a DNN, we assume the adversary has no information about the structure or parameters of the DNN and does not have access to any large training dataset. The adversary's only capability is to observe labels assigned by the DNN for chosen inputs in a manner analog to a cryptographic oracle. A successful attacking technique of this type is Su et al. (2019a) which is based on differential evolution. We can iteratively generate adversarial images to try to minimize the confidence (probability) of the network's classification.

The process starts with random modifications of a few pixels to generate adversarial examples. At each step, one feeds several adversarial images to the DNN and only observes the output as the probability of different classes. Examples that lowered the confidence of the true class will be kept to generate the next generation of adversaries. The next generation modifications are then evolved through a certain mutation scheme to generate new adversarial images. By repeating these steps for a few iterations, the adversarial modifications generate more and more misleading images. The last step returns an adversarial modification that reduces the confidence of the true class the most. This means that the confidence of the true class would be reduced so much that a new incorrect category now has the highest classification confidence. This technique is called untargeted attack since we don't specify the labels of the modified images.

In the following experiment, we modify one, three, and five pixels of images to generate adversarial examples. The mutation scheme we used for this experiment is as follows:

$$x_i^{l+1} = x_{r_1}^l + 0.5(x_{r_2}^l + x_{r_3}^l) \tag{6}$$

Where $r_1$, $r_2$, and $r_3$ are three non-equal random indices of the modifications at level $l$. $x_i^{l+1}$ will be an element of a new candidate modification.

To evaluate the effect of S-APL activation on the robustness of DNNs, we employ commonly-used architectures, namely, Lenet5, Network-in-Network (Lin et al., 2013), pure CNN (ours), and ResNet-18 (He et al., 2016). Each architecture is trained with ReLU, APL, and Swish activation functions and once with S-APL activation functions. The results in Table 2 show that Lenet5 and ResNet-18 architectures are 51% and 47.8% more resistant to back-box adversarial attacks. As authors in Zhao & Griffin (2016) theoretically proved, that one can be robust to adversarial examples with symmetric behavior.

After observing adversarial samples which are deceiving to both networks, we found that S-APL activated network still assigns high confidence to the true labels of the perturbed images. In other words, the difference between the confidence of the true label and the confidence of adversarially

Table 2: 100 images are randomly chosen from CIFAR-10 test set to generate adversarial examples. We attacked each architecture five times and report the average number of successful attacks. The maximum number of iterations for all attacks is set to 40.

| Model | Activation | one-pixel | three-pixels | five-pixels | $avg|Z_{true} - Z_{adv}|$ |
|-------|-----------|-----------|--------------|-------------|---------------------------|
| Lenet5 | ReLU | 61 | 88 | 92 | 0.721 |
|        | APL | 50 | 71 | 84 | 0.570 |
|        | Swish | 63 | 85 | 90 | 0.542 |
|        | S-APL | 29 | 58 | 67 | 0.401 |
| Net in Net | ReLU | 29 | 69 | 80 | 0.411 |
|            | APL | 24 | 50 | 69 | 0.400 |
|            | Swish | 29 | 71 | 77 | 0.446 |
|            | S-APL | 18 | 39 | 58 | 0.281 |
| Pure CNN | ReLU | 16 | 53 | 64 | 0.483 |
|          | APL | 12 | 44 | 58 | 0.301 |
|          | Swish | 14 | 51 | 67 | 0.312 |
|          | S-APL | 9 | 34 | 41 | 0.243 |
| ResNet18 | ReLU | 23 | 63 | 71 | 0.501 |
|          | APL | 17 | 45 | 60 | 0.413 |
|          | Swish | 22 | 65 | 70 | 0.541 |
|          | S-APL | 12 | 37 | 49 | 0.334 |

wrong label is much smaller in S-APL network than networks that are activated with ReLU, APL, or Swish. More precisely, we measure the average of $|Z(x')_{\text{true label}} - Z(x')_{\text{adversarial label}}|$ over all adversarial samples where each network is fooled. $Z(.)$ is the post-softmax output and $x'$ is the adversarial sample. The $avg(|Z(x')_{\text{true label}} - Z(x')_{\text{adversarial label}}|)$ for each model is also included in Table 2. Due to the reported averages, we can conclude that S-APL is much more robust against being adversarially deceived.

## 6.2 OPEN-BOX ADVERSARIAL ATTACK

To further explore the robustness of S-APL activated networks, in this section, we consider two of the popular benchmarks of open-box adversarial attacks: Fast Gradient Sign Method (FGSM) (Goodfellow et al., 2014) and CW attack (Carlini & Wagner, 2017). For both attacking strategy, we consider three different architectures and compare the rate of successful attacks for each of the networks with ReLU, APL, or Swich activations versus S-APL activation functions. The dataset and architectures are same as those were used for black-box adversarial attacks.

### 6.2.1 FGSM

Fast Gradient Sign Method generates an adversarial image $x'$ from the original image $x$ by maximizing the loss $L(x', y)$, where $y$ is the true label of the image $x$. This maximization problem is subjected to $||x - x'||_\infty \leq \epsilon$ where $\epsilon$ is considered as the *attack strength*. Approximating with the first term of Taylor expansion, we have:

$$L(x', y) = L(x, y) + \nabla_x L(x, y)^T.(x - x') \tag{7}$$

So the adversarial image $x'$ would be:

$$x' = x + \epsilon.sign(\nabla_x L(x, \theta)) \tag{8}$$

Where the $\epsilon$ is considered as the power of the attack. This form of FGSM is considered to be an untargeted attack where there is no pre-specified label $t$ that $x'$ should be classified as. This method only yields an adversary $x'$ which will not be classified as $y$.

To see how S-APL increases the robustness against FGSM attack, we employ three architectures all trained to classify CIFAR-10 dataset. For different range of $\epsilon$ the results are summarized in Table 3.

Table 3: Similar to the black-box attack experiment, 100 images are randomly chosen from CIFAR-10 test set to generate adversarial examples. For each architecture, four different $\epsilon$ is used to FGSM attack. For each model, four different activation namely, ReLU, APL, Swish, and S-APL are tested. The settings of the networks and activations are thoroughly explained in the Appendix.

| Model | Activation | $\epsilon = 0.02$ | $\epsilon = 0.04$ | $\epsilon = 0.06$ | $\epsilon = 0.08$ | $avg\|Z_{true} - Z_{adv}\|(\epsilon = 0.04)$ |
|---|---|---|---|---|---|---|
| Lenet5 | ReLU | 57 | 64 | 69 | 73 | 0.761 |
| | APL | 43 | 51 | 66 | 75 | 0.350 |
| | Swish | 54 | 59 | 71 | 74 | 0.619 |
| | S-APL | 36 | 44 | 62 | 68 | 0.298 |
| Pure CNN | ReLU | 35 | 36 | 38 | 42 | 0.721 |
| | APL | 30 | 34 | 38 | 42 | 0.509 |
| | Swish | 36 | 40 | 42 | 44 | 0.613 |
| | S-APL | 25 | 28 | 33 | 38 | 0.388 |
| ResNet18 | ReLU | 36 | 38 | 39 | 43 | 0.691 |
| | APL | 33 | 36 | 39 | 42 | 0.445 |
| | Swish | 36 | 39 | 40 | 42 | 0.511 |
| | S-APL | 27 | 29 | 35 | 40 | 0.391 |

Table 4: Similar to the FGSM attack experiment, 100 images are randomly chosen from CIFAR-10 test set to generate adversarial examples. For each model, four different activation namely, ReLU, APL, Swish, and S-APL are tested. The settings of the networks and activations are thoroughly explained in the Appendix.

| Model | Activation | | $avg\|Z_{true} - Z_{adv}\|$ |
|---|---|---|---|
| Lenet5 | ReLU | 83 | 0.871 |
| | APL | 79 | 0.551 |
| | Swish | 81 | 0.570 |
| | S-APL | 72 | 0.392 |
| Pure CNN | ReLU | 58 | 0.611 |
| | APL | 54 | 0.509 |
| | Swish | 56 | 0.6.31 |
| | S-APL | 47 | 0.365 |
| ResNet18 | ReLU | 49 | 0.689 |
| | APL | 44 | 0.461 |
| | Swish | 50 | 0.534 |
| | S-APL | 40 | 0.393 |

In the best case, for Lenet architecture and $\epsilon = 0.02$, S-APL activated network is 36% more robust than the ReLU activated one. The robustness also improves by 28% and 25% for pure CNN and ResNet-18 respectively. We can see the overall increase in the robustness over all networks and $\epsilon$ values. However, one can conclude that for smaller values of $\epsilon$ the robustness of S-APL activated network is higher in comparison to ReLU activated networks. As we increase the attack strength ($\epsilon$), the robustness of S-APL networks gets closer to the robustness of ReLU activated networks.

### 6.2.2 CW-L2

We showed that replacing ReLU with S-APL activation units can greatly improve the robustness of DNNs against FGSM attack. To further investigate the robustness that S-APL is adding to the network, we are using a more powerful open-box adversarial attack.

Another open-box adversarial attack which is generally more powerful than FGSM was introduced in Carlini & Wagner (2017). For a given pair of image $x$ and label $y$, this technique is trying to find the minimum perturbation $\delta$, so that the perturbed image $x'$ is classified as $t \neq y$. Considering $L_2$ norm as the goal of minimization, this perturbation minimization problem can be formulated as

follows:

$$\forall t \neq y, min||\delta||_2^2 \quad \text{subject to} \quad f(x + \delta) = t, \ \ x + \delta \in [0,1]^n \tag{9}$$

Or equally, to ease the satisfaction of equality, Equation 9 can be rephrased as $min||\delta||_2^2 + c.g(x+\delta)$ where $g(x) = max(max_{t\neq y}(logit(x)_t - logit(x)_y))$, $c$ is Lagrange multipliyer, and $logit(x)$ is the pre-softmax vector for the input $x$.

Here, we examine the robustness of S-APL activated networks against CW-L2 attack. As it is shown in the table 4, CW-L2 attack is more powerful than FGSM. Although the S-APL activated network is less robust to CW-L2 attack than to FGSM attack, it is still showing more robustness in comparison to ReLU, APL, and Swish activated networks. Similar to black-box attack, the metric $avg|Z_{true} - Z_{adv}|$ is calculated for successfully attacked images. The smaller values of $avg|Z_{true} - Z_{adv}|$ for S-APL activated networks in both open-box attacks shows that S-APL activated networks are more resistant to being fooled.

## 7    CONCLUSION AND FUTURE WORKS

We extend the idea of piecewise linear activation functions by introducing S-APL as a learnable and potentially symmetric activation function. We investigate the properties of S-APL and their evolution during training to understand how these functions accelerate learning and increase expressive power in DNNs. Finally, we show that S-APL networks show more robustness to adversarial attacks than ReLU networks. One can potentially design a hybrid defense method that combines other defense techniques and S-APL units.

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

## 8 APPENDIX

### 8.1 NUMBER OF HINGES AND THE SYMMETRY OF S-APL

In this section, we perform a variety of experiments to find the best setting for S-APL activation in terms of complexity and expressibility.

Before going to experiments, it is necessary to mention that we used batch normalization layer before every S-APL layer in our experiments. This means that the input of the S-APL activation is unit Gaussian. Relying on this fact and the experiments done in Agostinelli et al. (2014) we chose a fixed and symmetric set of hinges on $\pm\sigma, \pm2\sigma, \pm2.5\sigma, \pm3\sigma$, and ... . This will reduce the number of S-APL's parameters by a factor of 2 which makes it much less computationally expensive.

First, we assess the effect of $S$ on the performance of S-APL. Due to Theorem 3.1, greater $S$ values increase the expressive power of the S-APL which results in better fitting the training data. We tried $S \in [2, 3, 4, 5, 6]$, (which means 4, 6, 8, 10, and 12 hinges) with fixed hinges' locations on MNIST (LeCun & Cortes, 2010), CIFAR-10 (Krizhevsky et al.) and binary classification of IMDB review dataset (Maas et al., 2011) to cover MLP, convolution, and recurrent architectures respectively. Each network is trained with two types of S-APL activation; 1) A **shared S-APL**: a shared unit among all neurons of a layer and 2) An independent S-APL unit for each neuron of a layer. As it is summarized in Table 5, in all cases for $S \geq 4$ the performance is not decreasing significantly.

On the other hand, due to the increase in the number of parameters of S-APL, the unit becomes more computationally expensive. In Table 6 we compare the **per-epoch training run-time** for different number of hinges of a **shared S-APL**.

Table 5: Three classifications tasks are performed with five different numbers of hinges for S-APL activation. The number of additional parameters due to the use of S-APL and training loss are compared below. The MLP architecture consists of three layers each with 256, 64, and 32 units. For CIFAR-10, Lenet5 is employed and for the IMDB comment dataset, a network with one bi-LSTM layer followed by two MLPs with 32 and 16 units is used.

| # of hinges | 4 | 6 | 8 | 10 | 12 |
|---|---|---|---|---|---|
| **Error rate** | | | | | |
| MNIST | 1.73-1.61 | 1.70-1.57 | 1.61-1.57 | 1.53-1.53 | 1.55-1.59 |
| CIFAR-10 | 30.1-29.4 | 29.61-28.81 | 29.01-28.44 | 29.01-28.33 | 28.92-28.03 |
| IMDB | 6.11-6.35 | 5.87-5.91 | 5.41-6.12 | 5.47-6.32 | 5.41-7.24 |
| **# of additional params** | | | | | |
| MNIST | 12-1408 | 18-2112 | 24-2816 | 30-3520 | 36-4224 |
| CIFAR-10 | 16- >75k | 24->120k | 32->150k | 40->180k | 48->225k |
| IMDB | 12-384 | 18-576 | 24-768 | 30-960 | 36-1152 |

Table 6: Per-epoch training time is reported in seconds. The same number is calculated for Tanh and Maxout units (with six features). S-APL is faster than Maxout and is slightly slower than Tanh unit. All models are trained using NVIDIA TITAN V GPU with 12036MiB memory and 850MHz.

| Activation | **S-APL** | | | | | **Tanh** | **Maxout** |
|---|---|---|---|---|---|---|---|
| | 4 | 6 | 8 | 10 | 12 | | |
| MNIST | 5 | 7 | 10 | 12 | 13 | 6 | 12 |
| CIFAR-10 | 18 | 21 | 24 | 27 | 30 | 21 | 31 |
| IMDB | 11 | 14 | 14 | 15 | 16 | 12 | 18 |

For small values of $S$, we can see that S-APL is comparable to an exponential activation unit such as Tanh, and much faster than heavier activation such as Maxout.

As one can conclude from both Table 5 and 6, there is a trade-off between the complexity and expressibility. We believe that $S = 4$, or equally 8 hinges, is the best choice for the number of hinges where the hinges' locations are symmetrically fixed.

## 8.2 Experiments' Details and Statistical Significance

In this section, we explain the experimental conditions and all the parameters used for each experiment. Also, in order to make the results of Table 1 more interpretable, we perform a t-test (Kim, 2015) on all the error rates achieved in that experiment.

**In section 4**, the experiment corresponding to Table 1 are performed using six different architectures. Lenet 5 is used as it was introduced in LeCun et al. (1998). It has two convolution layers followed by two MLPs that are connected to a softmax layer.

The MLP architecture which is only used for classifying MNIST dataset has three fully connected feed-forward layers with 258, 128, and 10 hidden units. A dropout of rate $0.3$ is added in between each layer. We use the batch size of 32 and we train the MLP network for 50 epochs.

Pure CNN architecture which is only taking advantage of convolutional layers, was introduced in Springenberg et al. (2014). All the specifications, including kernel size, stride, learning rate, batch size, and the number of epochs are taken from the main article.

ResNet18 architecture as a variant of residual networks (He et al., 2016) is taken from Napoletano et al. (2018) which has five residual convolution blocks, one MLP, and a softmax layer. The size of the convolution kernels, strides, padding, pooling size, pooling operation, learning rate, batch size, and epochs are taken as the defaults mentioned in Napoletano et al. (2018).

EffectiveNets was introduced in Tan & Le (2019) and is used in our experiments exactly the way in was originally introduced. It is necessary to mention that we use $B0$ variant of EffectiveNets.

Lastly, Net in Net architecture which is using an MLP instead of a fixed nonlinear transformation is taken from Agostinelli et al. (2014).

It is worth mentioning that in section 4, all the experiments were performed five times. To reduced the effect of noise and randomness, the average of the five times is reported. In Tabel 7, we show the statistical significance of the experiments performed in section 4. Since each number is the average of five experiments, we are able to perform a t-test and provide p-values and statistical significance for each individual experiment. As one can see in Table 7, most of the numbers of Table 1 are statistically significant.

Table 7: The best activation among ReLU, leaky-ReLU, PReLU, tanh, sigmoid, ELU, maxout (nine features), Swish: $x.sigmoid(\beta x)$ is chosen by the minimum average of the error rate. Then the significance of the comparison between the best network and S-APL activated network is calculated through a t-test. The p-vales for each comparison is provided below.

| Activation | MNIST | CIFAR-10 | | CIFAR-100 | |
|---|---|---|---|---|---|
| | - | - | D-A | - | D-A |
| Lenet5 (PReLU vs S-APL) | 0.061 | 0.041 | 0.049 | 0.054 | 0.062 |
| MLP (Swish vs S-APL) | 0.073 | | | | |
| pure CNN (maxout vs S-APL) | | 0.043 | 0.057 | 0.072 | 0.068 |
| ResNet-18 (PReLU vs S-APL) | | 0.029 | 0.031 | 0.049 | 0.040 |
| EffectiveNet B0 (ReLU vs S-APL) | | 0.052 | 0.042 | 0.055 | 0.059 |
| Net-in-Net(ReLU vs S-APL) | | 0.012 | 0.019 | 0.041 | 0.050 |

**In section 5**, we use two architectures to visualize S-APL shapes at different stages of the training process. For Figure 1 and Figure 2 we use the same MLP and Lenet 5 architectures as described above.

**In section 6**, we start by a tSNE visualization of 100 random samples of frogs and ships from the cifar10 test set. The tSNE mapping is performed using a learning rate of 30 and a perplexity of 40.

For the black-box adversarial attack experiments, each network is attacked five times and the reported number is the average of successful modifications in five different attacks. Attacks are done using the maximum number of iteration to be 40 and the pop size to be 400.

As for the open-box attacks, for both FGSM and CW-L2 attack, we employ the implementation and default hyper-parameters in Rauber et al. (2017).

Lastly, **for activation function of networks in section 6**, ReLU ($y = x$ for $x > 0, 0$ otherwise ), APL ($S = 5$, with fixed hinges on $0, \pm 1,$ and $\pm 2$), Swish ($y = x.sigmoid(\beta x)$ with $\beta = 0.2$), and S-APL (with the configurations mentioned in the previous section) are used.

