# OpenReview forum: "Symmetric-APL Activations: Training Insights and Robustness to Adversarial Attacks"
_ICLR.cc/2020/Conference — Reject_

### Official Review · AnonReviewer4 · 2019-10-21
**Official Blind Review #4**

**Rating:** 1

**Review:**

This paper proposes the S-APL (Symmetric Adaptive Piecewise Linear) activation function, based on the APL activation function proposed by (Agostinelli et al, 2014). This activation function is constructed as a piecewise linear function that is learned concurrently with training, and, in the case of S-APL, the activation function is forced to be symmetric. S-APL is claimed to help both with trainability and robustness of neural networks to adversarial examples.

Overall, the idea is clearly presented, but appears to have many critical flaws, enumerated below:
1. It is unclear what the motivation for the symmetry is upon first reading of the paper, i.e., Section 3.1 starts by saying "To overcome the shortcomings of APL," but up to this point I cannot find any exposition that explains what these shortcomings are---the beginning of section 3 just presents the formulation of APL and does not discuss its advantages/disadvantages.

2. The experimental results concerning network training are very hard to interpret, as they lack error bars, confidence intervals, and many critical experimental details (e.g. how many networks were trained, the hyper parameters for training, etc.) For these results to be interpretable, the authors should include a detailed description of the environment under which the experiments were performed, and present confidence intervals which demonstrate the significance of the improvement attained by S-APL. (As it stands, it is hard to tell whether, e.g., a 0.1% improvement on CIFAR-10 should be considered significant.)

3. The adversarial evaluation section should be substantially revised to address several important flaws:
(a) The evaluation for black-box attacks is done in very non-standard threat models (e.g. label-only 3-pixel black-box attacks) that do not seem to be relevant to the black-box robustness of a system. Even under these threat models, the authors should also use more powerful label-based black-box attacks, such as the BoundaryAttack [1] or the label-only attacks in [2] or [3].
(b) The avg(|Z_{true} - Z_{adv}|) has an absolute value around Z_{true} - Z_{adv}, which means that failed adversarial attacks actually contribute positively to the score, which severely limits its usefulness for judging adversarial robustness.
(c) The open-box (white-box) setting only uses FGSM to evaluate the robustness, which is known to be a weak attack [4] and is recommended against for evaluating adversarial robustness [5]. The methods should be evaluated with PGD or CW-attacks to better judge robustness.
(d) Overall, the results presented are not sufficient to evaluate the adversarial robustness of the S-APL activation. A first step towards remedying this would be to follow the evaluation protocol suggestions outlined in [5].

The idea does seem promising, however, and the paper could be substantially improved by including the necessary introduction, evaluation protocols, and experimental details. However, this would require a substantial change to the paper, and thus my recommendation for now is to reject.

[1] https://arxiv.org/abs/1712.04248
[2] https://arxiv.org/abs/1804.08598
[3] https://arxiv.org/abs/1807.04457
[4] https://arxiv.org/abs/1802.00420
[5] https://arxiv.org/abs/1902.06705

**Experience Assessment:**

I have published in this field for several years.

**Review Assessment: Checking Correctness Of Derivations And Theory:**

N/A

**Review Assessment: Checking Correctness Of Experiments:**

I carefully checked the experiments.

**Review Assessment: Thoroughness In Paper Reading:**

I read the paper thoroughly.

---

> ### Author Response · Authors · 2019-11-15
> **Authors' Responses to Review #4**
>
> We thank the reviewer for the effective review and constructive feedback! We are glad that the reviewer found the idea promising.
>
> 1. Reviewer 4 makes a great point that in section 3.1, it is hard to find the disadvantages of APL which results in ambiguity in the motivation. We agree with this point so we revised the writing of section 3.1 to shed light on the disadvantages of the normal APL. Then, in section 3.2 we enumerate the advantages of S-APL over APL so the reader can understand the motivation behind proposing S-APL.
> The added exposition in section 3.1 explaining the shortcomings of APL can be summarized as follows:
> APL units are not zero-centered as they can have a non-zero output for an input of zero. This behavior provides no apparent beneficial purpose and is not present for S-APLs. Furthermore, APLs can only represent piecewise linear functions whose output g(x) = x for x > u, for some scalar u. This significantly restricts the class of piecewise linear functions that APLs can express. S-APLs, on the other hand, do not have this restriction, broadening the scope of functions they can approximate. We have included this in the updated text.
>
> 2. We appreciate reviewer 4 suggested that performing a statistical significance test makes the results more interpretable. In the general revisions, we addressed this issue by adding a new section in the appendix, under which we performed statistical tests to provide t-values and p-values. We hope that the provided analysis had brought more interpretability for the results of table 1. As one can see almost all the numbers are statistically significant.
>
> 3. Due to the comment from R4 on adversarial attack experiments, we have added CW-L2 as another open-box attack which is generally much more powerful than FGSM. As the experiments show, although the S-APL activated network is less robust to CW-L2 attack than FGSM attack, it is still showing more robustness in comparison to ReLU, APL, and Swish activated networks.
> Other than that, we have added APL and Swish activated networks to all the experiments in section 6.
> R #4 also brought that since failed adversarial attacks contribute positively to the score, avg(|Z_{true} - Z_{adv}|) is useless for judging adversarial robustness. We think it is necessary to mention that as we stated in section 6, the metric avg(|Z_{true} - Z_{adv}|) is the average over adversarial images for which the network is fooled. Considering this clarification, we think the metric can reasonably represent the robustness to attacks.

---

### Official Review · AnonReviewer1 · 2019-10-25
**Official Blind Review #1**

**Rating:** 3

**Review:**

In this paper, a new activation function, i.e. S-APL is proposed for deep neural networks. It is an extension of the APL activation, but is symmetric w.r.t. x-axis. It also has more linear pieces (actually S pieces, where S can be arbitrarily large) than the existing activation functions like ReLU. Experimental results show that S-APL can be on par with or slightly better than the existing activation functions on MNIST/CIFAR-10/CIFAR-100 datasets with various networks. The authors also show that neural networks with the proposed activation can be more robust to adversarial attacks.

First of all, the activation function is much more complicated than the existing ones, as it has to determine the parameter S and the hinge positions. However, the gain is marginal as shown in Table 1. Besides, the authors never tell how to choose S and the hinge positions.

Secondly, the neural networks used in the experiments are quite outdated. And the error rates shown in Table 1 are far away from state-of-the-art. Why don't you choose a latest network such as ResNet/DenseNet/EfficientNet and replace the activation with S-APL? The results could be more convincing.

I am not an expert in adversarial attack. But is there any intuition why a complicated activation function is more robust to adversarial attack? Again, most of the models used in Table 2 are quite old (Lenet5, Net in Net, CNN).

In a word, the proposed activation function is unnecessarily complicated and the gain is not justified with the latest models and not significant enough to convince people to adopt it.


**Experience Assessment:**

I have read many papers in this area.

**Review Assessment: Checking Correctness Of Derivations And Theory:**

I assessed the sensibility of the derivations and theory.

**Review Assessment: Checking Correctness Of Experiments:**

I assessed the sensibility of the experiments.

**Review Assessment: Thoroughness In Paper Reading:**

I made a quick assessment of this paper.

---

> ### Author Response · Authors · 2019-11-15
> **Authors' Responses to Review #1**
>
> Thank you for the detailed reviews. In the general post, we have addressed your chief concern regarding the use of more up to date architecture. We sincerely hope R1 can revisit the rating in light of our revision and response.
>
> 1. On the complexity of the S-PAL and marginal gain:
> We thank R1 for this comment so we can provide more clarification on the benefits of S-APL.
> First, the value of S and the hinges’ positions do not need to be tuned for different applications. We use the same value of S and the same hinge positions for all experiments. Since we use batch-normalization before the activation function, the hinge position can be set at intuitive locations that correspond to standard deviations. We set the hinge locations at 0, 1, 2, and 2.5 for both the positive and negative sides of the activation function. We add details for how we choose these values in the appendix. In terms of the complexity related to the number of parameters, the S-APL only adds 8 parameters per layer.
> In terms of the benefits of using S-APL, we aimed to show that the S-APL unit is the only learned activation function that both improves classification accuracy as well as makes the network more robust to adversarial attacks. Although the improvement in classification tasks seems marginal, the competition in this area is seeking 0.01 improvements.
>
> 2. On the use of more updated networks, we agree with the reviewer and we have added ResNet-18 and EffectiveNet to Table 1. The results show that S-APL improves classification performance in those cases as well. Both of these networks are proposed recently are they are commonly used networks in the community.
>
> 3. R1 also brought the important question, “is there any intuition why a complicated activation function is more robust to adversarial attack?”
> There are a considerable number of research focusing on improving activation functions as a defense mechanism against adversarial attacks such as [1] and [2]. The authors in [3] also provided a theoretical justification for the important role of activation function as a reason for the vulnerability of DNNs against adversarial foolings. They showed that vulnerability is caused by a failure to suppress unusual signals within network layers. As a remedy, they propose the use of Symmetric Activation Functions (i.e. even functions) in non-linear signal transducer units. These units suppress signals of exceptional magnitude. THey mathematically proved that symmetric networks can also perform classification tasks to arbitrary precision. On the other hand, the learnable activation has shown great superiority over fixed activation in the past few years. Our paper is taking advantage of both ideas of learning the activation and symmetric shape so it can be beneficial in both aspects.
>
> [1] Rakin, Adnan Siraj, et al. "Defend deep neural networks against adversarial examples via fixed anddynamic quantized activation functions." arXiv preprint arXiv:1807.06714 (2018).
> [2] Wang, Bao, et al. "Adversarial defense via data dependent activation function and total variation minimization." arXiv preprint arXiv:1809.08516 (2018).
> [3] Zhao, Qiyang, and Lewis D. Griffin. "Suppressing the unusual: towards robust cnns using symmetric activation functions." arXiv preprint arXiv:1603.05145 (2016).

---

### Official Review · AnonReviewer2 · 2019-10-30
**Official Blind Review #2**

**Rating:** 6

**Review:**

This paper proposes a learnable piece-wise linear activation unit whose hinges are placed symmetrically. It gives a proof on the universality of the proposed unit on a certain condition. The superiority of the method is empirically shown. The change of the activation during training is analyzed and insight on the behavior is provided. The robustness to adversarial attacks is also empirically examined.

This paper discusses a very basic component of neural network models: activation function. Thus, it should be of interest to many researchers. The proposed method is simple and seems easy to use in real settings. A number of experiments are conducted to validate the method and the results look promising. The experiments in Section 5 is particularly interesting. It might give some hints for the following studies.

However, there are several things to be addressed for acceptance.

1) What is actually proposed is not very clear.

S-APL is formulated in Equation 2. However, there are some discussion after that which changes or restricts the equation. For example, it seems that b_i^s^+ = b_i^s^- is assumed throughout the paper. In that case, it should be just reflected in Equation 2. In the third paragraph of Section 3.2, it is mentioned that h_i^s(x) = h_i^s(-x) with b_i^s^+ = b_i^s^-. However, it should also assume that a^s^+ = a^s^-. From the experiments. apparently, a^s^+ = a^s^- is not assumed. It seems that the method has symmetry only for the hinge locations.

In the first paragraph of Section 3.2, it is implied that parameters are shared across layers. It is not very clear what is shared and what is not. Please make that part clear. It will make it easier to understand the experimental settings, too.

2) Theorem 3.1 does not seem to prove the approximation ability of S-APL.

It is clear that g(x, S) can represent arbitrary h(x, S), but I am not sure if it is clear that h(x, S) can represent arbitrary g(x, S). It should also depend on the conditions on a^s^+, a^s^-, b_i^s^+, b_i^s^-. I think it needs to prove that h(x, S) can approximate arbitrary piecewise linear function (i.e., g(x, S)) if you want to prove the approximation ability of h(x, S).

Equation 4 seems to assume that all intervals are the same (i.e., ∀i, B_i - A_i = (B-A) / S). It should be stated explicitly. This relates to the problem 1).

I may not understand some important aspect. I am happy to be corrected.

3) Experimental conditions are not clear.

Please cite the papers which describe the architecture of the models used in the experiments. The effectiveness of the proposed method should depend on the network architecture and it is importable to be able to see the details of the models.

4) On the sensitivity of optimization on the initial value.

It is interesting to see that "fixed trained S-APL" is not comparable with "S-APL positive". If the hypothesis in the paper is correct, it is natural to assume that "fixed trained S-APL" also has some issue on training. It would be interesting to see experimental results with "initialized with trained-S-APL" and "S-APL positive with non-zero initial value".  It is a bit weird to observe that "S-APL positive" never becomes non-zero for x < 0.

5) Comparison results with other activation units in Section 6.

The proposed method is compared only with ReLU. It is important to see comparisons with other activations such as the plain APL.


Some other minor comments:

It is quite interesting that objects are actually modified for adversarial attack for the proposed method in Figure 5. It would be interesting to have some consideration on it.

**Experience Assessment:**

I have read many papers in this area.

**Review Assessment: Checking Correctness Of Derivations And Theory:**

I carefully checked the derivations and theory.

**Review Assessment: Checking Correctness Of Experiments:**

I carefully checked the experiments.

**Review Assessment: Thoroughness In Paper Reading:**

I read the paper at least twice and used my best judgement in assessing the paper.

---

> ### Author Response · Authors · 2019-11-15
> **Authors' Responses to Review #2**
>
> We thank the reviewer for the detailed reviews and constructive feedback! We apologize for the somewhat delayed response; it took us time to run additional experiments and add more careful analysis so that we can present an improved and more polished paper to everyone. We appreciate your understanding.
>
> In the following, we address the main concerns of the reviewer in the order we received.
>
>
> 1. The reviewer makes a great observation that the S-APL is not clearly proposed and we acknowledge that the proposed formulation for S-APL and its following restrictions were not aligned with the assumption of symmetry. For that purpose, both a^i_+ = a^i_- and b^i_+ = b^i_- have to be satisfied.
> The restrictions mentioned are to demonstrate that S-APLs can hypothetically take on a symmetric shape. In our experiments, we see that the final shapes are approximately symmetric (i.e. in Figure 2 and Figure 3). We have updated the language used in the paper to make this clear.
> The reviewer also brought the point that “parameters are shared across layers” is not a clear statement. We have updated that paragraph’s language by “S-APL shares the variables a_{+}^s, b_{+}^s, a_{-}^s, and b_{-}^s among all the neurons of a layer (e.i. h_i(x, S) does not depend on i)”
>
> 2. We appreciate the useful comment “Theorem 3.1 does not seem to prove the approximation ability of S-APL” from the reviewer. We agree that we needed to prove that h(x, S) can approximate arbitrary piecewise linear function (i.e., g(x, S)). This could be done by setting the a^i and b^i s to mimic g(x, S), however, we found it more beneficial to provide a more straightforward proof. We fixed this issue by furnishing a new proof which directly shows that S-APL can approximate any M-Lipschitz continuous functions in an interval of real numbers. The new proof is provided as Theorem 3.1 is in the updated version.
>
> 3. In regards to another comment “sensitivity of optimization on the initial value”, we have added the loss trajectory of the S-APL initialized with the final shape of a trained S-APL. As one can see, this new initial state ends up with a lower loss than fixed trained S-APL. However, the ReLU init S-APL is still outperforming other initial states. This observation is strengthening our hypothesis of the two required stages of accelerating gradient and improving expressibility.
>
> 4. Due to a great suggestion from the reviewer about comparing the robustness of networks equipped with activations other than ReLU and S-APL, we added Swish and plain APL to all the experiments of section 6. All the additional experiments show that S-APL has higher robustness to other activations.

---

### Author Response · Authors · 2019-11-15
**Paper update overview**

We sincerely appreciate all the reviews, they give high-quality comments on our paper with a lot of constructive feedback. In the revised paper, we did our best to address the concerns and suggestions to strengthen our paper. We sincerely hope reviewers revisit the rating in light of our revision and response. The following summarizes our major revisions. Please see our rebuttal for the detailed discussion.

General revisions:

1. Based on the helpful comment of Reviewer 2 on the ambiguity of the experimental conditions, we have added a new subsection “Experiments' Details and Statistical Significance” in the appendix where we specified all the details, conditions, and hyper-parameters of the experiments. Adding this section can be super useful for the readers and enable them to reproduce the experiments’ results.

2. Due to another insightful comment from Reviewer 2 on comparing the robustness of S-APL with more activation functions, we have added two of the recent and successful activations namely, APL and Swish to all the experiments of section 6. These additional experiments further demonstrate the superiority of the S-APL activated networks on the robustness to adversarial attacks.

3. R1 makes a great point that it is not clear how the main hyper-parameter of S-APL, “S” is chosen. To address this, we have attached a subsection in the appendix titled “Number of Hinges and the Symmetry of S-APL”. Within this section, we empirically showed how to choose the parameter “S” and how to reduce the complexity of the activation by using a shared S-APL for all the neurons of one layer.

4. A great comment by R1, stated that the architectures used in our experiments are not the most updated ones and adding more recent networks would make it more convincing for the community to adopt S-APL. We appreciate this helpful comment. In regards to that, we have added ResNet-18 and EfficientNet as two of the up to date and highly used network in section 4.

5. Reviewer 4 made a clever point about the interpretability of the experiments in section 4. We have performed several statistical tests to provide p-values and show the statistical significance of the numbers presented in section 4. Within the newly added section in the appendix, we have calculated the statistical significance of the results for each of the networks in Table 1.

---

### Decision · Program_Chairs · 2019-12-19

**Decision:**

Reject

**Comment:**

This work presents a learnable activation function based on adaptive piecewise linear (APL) units. Specifically, it extends APL to the symmetric form. The authors argue that S-APL activations can lead networks that are more robust to adversarial attacks. They present an empirical evaluation to prove the latter claim. However, the significance of these empirical results were not clear due to non-standard threat models used in black-box setting and the weak attacks used in open-box setting. The authors revised the submission and addressed some of the concerns the reviewers had. This effort was greatly appreciated by the reviewers. However, the issues related to the significance of robustness results remained unclear even after the revision. In particular, as pointed by R4, some of the revisions seem to be incomplete (Table 4). Also, the concern R4 had initially raised about non-standard black-box attacks was not addressed. Finally, some experimental details are still missing. While the revision indeed a great step, the adversarial experiments more clear and use more standard setup be convincing.